# Genetic Analysis of Hexaploid Wheat (*Triticum aestivum* L.) Using the Complete Sequencing of Chloroplast DNA and Haplotype Analysis of the *Wknox1* Gene

**DOI:** 10.3390/ijms222312723

**Published:** 2021-11-24

**Authors:** Mari Gogniashvili, Yoshihiro Matsuoka, Tengiz Beridze

**Affiliations:** 1Institute of Molecular Genetics, Agricultural University of Georgia, #240 D. Aghmashenebeli Av, Tbilisi 0131, Georgia; t.beridze@agruni.edu.ge; 2Graduate School of Agricultural Science, Kobe University, Rokko, Nada, Kobe 657-8501, Japan; pinehill@port.kobe-u.ac.jp

**Keywords:** *Triticum aestivum* L., chloroplast DNA, sequencing, Illumina, SNP, *Wknox1* gene

## Abstract

The aim of the presented study is a genetic characterization of the hexaploid wheat *Triticum aestivum* L. Two approaches were used for the genealogical study of hexaploid wheats—the complete sequencing of chloroplast DNA and PCR-based haplotype analysis of the fourth intron of *Wknox1d* and of the fifth-to-sixth-exon region of *Wknox1b.* The complete chloroplast DNA sequences of 13 hexaploid wheat samples were determined: Free-threshing—*T. aestivum* subsp. *aestivum*, one sample; *T. aestivum* subsp. *compactum*, two samples; *T. aestivum* subsp. *sphaerococcum*, one sample; *T. aestivum* subsp. *carthlicoides,* four samples. Hulled—*T. aestivum* subsp. *spelta*, three samples; *T. aestivum* subsp. *vavilovii* jakubz*.,* two samples. The comparative analysis of complete cpDNA sequences of 20 hexaploid wheat samples (13 samples in this article plus 7 samples sequenced in this laboratory in 2018) was carried out. PCR-based haplotype analysis of the fourth intron of *Wknox1d* and of the fifth-to-sixth exon region of *Wknox1b* of all 20 hexaploid wheat samples was carried out. The 20 hexaploid wheat samples (13 samples in this article plus 7 samples in 2018) can be divided into two groups—*T. aestivum* subsp. *spelta,* three samples and *T. aestivum* subsp. *vavilovii* collected in Armenia, and the remaining 16 samples, including *T. aestivum* subsp. *vavilovii* collected in Europe (Sweden). If we take the cpDNA of Chinese Spring as a reference, 25 SNPs can be identified. Furthermore, 13–14 SNPs can be identified in *T. aestivum* subsp. *spelta* and subsp. *vavilovii* (Vav1). In the other samples up to 11 SNPs were detected. 22 SNPs are found in the intergenic regions, 2 found in introns, and 10 SNPs were found in the genes, of which seven are synonymous. PCR-based haplotype analysis of the fourth intron of *Wknox1d* and the fifth-to-sixth-exon region of *Wknox1b* provides an opportunity to make an assumption that hexaploid wheats *T. aestivum* subsp. *macha* var. *palaeocolchicum* and var. *letshckumicum* differ from other macha samples by the absence of a 42 bp insertion in the fourth intron of *Wknox1d*. One possible explanation for this observation would be that two *Aegilops tauschii* Coss. (A) and (B) participated in the formation of hexaploids through the D genome: *Ae. tauschii* (A)—macha (1–5, 7, 8, 10–12), and *Ae. tauschii* (B)—macha M6, M9, *T. aestivum* subsp. *aestivum* cv. ‘Chinese Spring’ and cv. ‘Red Doly’.

## 1. Introduction

Wheat is the leading grain crop in the world. It originated in the Fertile Crescent approximately 10,000 years ago and has since spread worldwide. There are two biological species of hexaploidy wheat—*T. aestivum*, genome BBA^u^A^u^DD and *Triticum zhukovskyi* Menabde & Ericz., GGA^u^A^u^A^m^A^m^. *T. zhukovskyi* and its predecessors (*Triticum monococcum* L. and *Triticum timopheevii* (Zhuk.) Zhuk.) form a separate lineage irrelevant to the evolution of the principal wheat lineage, which is formed by *T. aestivum* and its predecessors *Aegilops tauschii* Coss. And *Triticum turgidum* L. [1].

The *T. aestivum* lineage is divided into the domesticated, hulled lineage and the free-threshing lineage. The free-threshing lineage includes *T. aestivum* subsp. *Aestivum*, *T. aestivum* subsp. *Compactum* (Host) Mackey, *T. aestivum* subsp. *Sphaerococcum* (Percival) Mackey, and *T. aestivum* subsp. *carthlicoides* nom. nud. (Table 1 and Table 2).

Hexaploid wheat *T. aestivum* subsp. *carthlicoides* was found by Kuckuck [3] near the border of Turkey, Armenia and West Georgia. This hexaploid wheat showed the subsp. *carthlicum*-like spike morphology.

The hulled lineage includes spelt wheat *T. aestivum* subsp. *spelta* Thell., macha *T. aestivum* subsp. *macha* (Dekapr. and Menabde) Mackey and *T. aestivum* subsp. *vavilovii* Jakubz. In hulled wheats, glumes tenaciously enclose seeds, and strong mechanical force is needed to liberate seeds from glumes during threshing. 

According to Dekaprelevich, Georgia is characterized by the largest number of cultivated wheat species in the world; altogether, 12 species of are found here. Only three narrowly endemic species, *Triticum abyssinicum* Vav.*, T. sphaerococcum* and *T. spelta,* are absent [4].

South Caucasus (notably Georgia) and its earlier residents played an important role in wheat formation. In total, 17 domesticated species and subspecies of *Triticum* are known. Georgian endemic wheat species include one *Triticum* species and four subspecies [5,6,7]: *Triticum turgidum* subsp. *palaeocolchicum* (Menabde) A. Love*Triticum turgidum* subsp. *carthlicum* (Nevski) A. Love*Triticum timopheevii* subsp. *zhukovskyi* (Menabde & Ericzjan) L. B. Cai*Triticum zhukovskyi* Menabde & Ericzjan*Triticum aestivum* subsp. *macha* (Dekapr. & Menabde) McKey

All these cultivated species and subspecies have been found only in West Georgia, except subsp. *carthlicum* which has been distributed in East Georgia as well. All these species and subspecies grew in the territory of Georgia until the middle of the last century.

The “Wheat Enigma” was a term for the observation that wild predecessors of five Georgian endemic wheat subspecies are found in Fertile Crescent, quite far from the South Caucasus [8,9]. One possibility to explain the “Wheat Enigma” is that speakers of ProtoGeorgian language could have moved to Mesopotamia after migration from Africa to the Arabian Peninsula, where wheat was domesticated. Furthermore, they could have migrated to South Caucasus together with domesticated wheat subspecies [9,10].

The examination of genealogical data provides insights into the evolutionary history of a species. The wide application of gene genealogies for evolutional studies in plants involves identifying DNA sequences with levels of ordered variation within chloroplast, mitochondrial, or nuclear genomes [11]. Traditionally, extranuclear DNA, such as chloroplast DNA (cpDNA), has been considered as an effective tool for genealogic studies [12,13,14]. The sequences of wheat plasmons B and G (complete cpDNA sequences) of the genus *Triticum* were determined in our laboratory [8,15,16]. Plasmon B is detected in polyploid species—*Triticum turgidum* and *T. aestivum.* Plasmon of *T. zhukovskyi* belong to the G type. 

Another effective tool for gene genealogy studies for *Triticum* species is the three homoeologous loci of wheat *Wknox1* gene, functioning at shoot apical meristems (SAM) [17]. A comparative study of the three *Wknox1* genomic sequences revealed accumulation of numerous mutations, particularly in the fourth intron and the 5′-upstream region. Later, Takumi a. Morimoto [18] reported the discovery of a new allele for the fifth-to-sixth exon region of the *Wknox1b KNOTTED1*-type homeobox gene in a common wheat subspecies (*T. aestivum* subsp. *carthlicoides*).

The purpose of the present investigation was to carry out the comparative analysis of complete cpDNA sequences of 20 hexaploid wheat accessions (*T. aestivum* subsp. *aestivum*, two samples; *T. aestivum* subsp. *compactum*, two samples; *T. aestivum* subsp. *sphaerococcum*, one sample; *T. aestivum* subsp. *carthlikoides,* four samples; *T. aestivum* subsp. *spelta*, three samples; *T. aestivum* subsp. *macha*, six samples; *T. aestivum* subsp. *vavilovii**,* two samples.

The second aim of the present investigation was to carry out a PCR-based haplotype analysis of the fourth intron of *Wknox1d* and of the fifth-to-sixth exon region of *Wknox1b* of all 20 hexaploid wheat samples.

## 2. Results

### 2.1. Complete cpDNA Sequence of Hexaploid Wheats

For comparative analysis of chloroplast DNA of hexaploid wheats, 20 hexaploid wheat samples were selected. CpDNA sequences from 13 of them were sequenced in this study, and the remaining 7 were sequenced earlier [15] (Table 3). 

The samples were structured as follows: free-threshing—*T. aestivum* subsp. *aestivum*, two samples—Chinese Spring and Red Doly; *T. aestivum* subsp. compactum—two samples; *T. aestivum* subsp. *sphaerococcum*—one sample; *T. aestivum* subsp. *carthlicoides*—four samples; Hulled—*T. aestivum* subsp. *spelta*—three samples; *T. aestivum* subsp. *vavilovii* two samples; *T. aestivum* subsp. *macha*—six samples.

To illustrate the evolutionary relationship among the studied cultivars, a phylogenetic tree was constructed based on complete nucleotide sequences of cpDNA of 20 hexaploid wheat samples (Figure 1). 

If we take the cpDNA of Chinese Spring as a reference, 25 SNPs can be identified. Furthermore, 13–14 SNPs can be identified in *T. aestivum* subsp. *Spelta* and subsp. *Vavilovii* (Vav1) (collected in Armenia) (Table 4). In the other samples, up to 11 SNPs were detected. In total, 22 SNPs are found in the intergenic regions, 2 were found in introns, and 10 SNPs were found in the genes, of which seven are synonymous and do not alter the amino acids. Indels specific for 20 cpDNA are given in Table 5.

One 35bp insertion and three inversions (56bp, 58bp and 25bp in length) have been identified in *Triticum aestivum* subsp. *spelta* (Splt1, Splt2, Splt3) and *T. aestivum* subsp. *vavilovii* (Vav2) samples (Table 6). One 38bp inversion (TCGGCTCAATCTTTTTTTTCTAAAAAAGATTGAGCCGA) with a 4bp loop has been identified in *T. aestivum* subsp. *vavilovii* (Vav1) (collected in Armenia) in the position 56,120–56,157 (intergenic *rbcL—psaI*). 

This section may be divided into subheadings. It should provide a concise and precise description of the experimental results, their interpretation, as well as the experimental conclusions that can be drawn.

### 2.2. PCR Analysis of Three Homoeologous Loci of Wheat Wknox1 Gene

#### 2.2.1. *Wknox1d* Fourth Intron Region

In the present investigation, the genomic sequences of the *Wknox1d* fourth intron regions were amplified by PCR using the primer pair of Takumi a. Morimoto [18] (Table 7). A common wheat, *T. aestivum* subsp. *aestivum* cv. ‘Chinese Spring’ (CS), and a durum wheat, *T. turgidum* subsp. *durum* cv. ‘Langdon’ (Ldn), were used for PCR as control and the amplified DNA fragments were visualized on an agarose gel. In the fourth intron of *Wknox1d* in common wheat, a 122-bp MITE insertion has been reported [17]. The MITE-containing band (411 bp) was missing in all tetraploid wheat accessions and was observed in all subspecies of common wheat. In the case of subsp. *macha* (11 samples) the 453 bp band was observed. In the case of two macha samples (M6 and M9) the MITE-containing band typical for other hexaploid subspecies (411 bp) were detected.

In 2% agarose-gel, the PCR-amplified, fourth intron of the *Wknox1d* DNA region of hexaploid wheat gives four bands: N1 (280 bp), N2 (375 bp), N3 (411 bp), and N4 (453). Bands N3 and N4 were cut out and sequenced. In the case of band N4 (453), a 42 bp insertion was detected in the position 7009—AGTTTGCACACCTGAACATTTTGCATTATGTTCGGGAGCCTA (Figure 2).

#### 2.2.2. Fifth-to-Sixth Exon Region of *Wknox1b*

PCR-based haplotype analysis of the fifth-to-sixth exon region of *Wknox1b* showed that the 157-bp MITE inserted band (588 bp) is present in *T. turgidum* subsp. *durum* cv. ‘Langdon’; CS; Red Doly (Table 8). This band is absent in tetraploid *T. turgidum* subsp. *carthlicum,*
*T. aestivum* L. subsp*. carthlicoides,*
*T. aestivum* subsp. *macha* (M1–M13).

Hexaploid wheats with matching values of the fourth intron of *Wknox1d* and the fifth-to-sixth-exon region of *Wknox1b* are given in Table 9.

## 3. Discussion

It is widely believed that the birthplace of *T. aestivum* is in a region between Transcaucasia and southwestern Caspian Iran [19]. The first step in the evolution of cultivated wheat was the formation in northern Mesopotamia of a tetraploid species with an A^u^A^u^BB genome [20]. Approximately 7000 years ago, the hexaploid bread wheat *T. aestivum* L. (BBA^u^A^u^DD) arose in southwestern Caspian Iran and Transcaucasia by allopolyploidization of the cultivated Emmer wheat *Triticum dicoccum* Schrank with the Caucasian *Ae. tauschii* [19,21,22,23,24].

It should be noted that *Ae. tauschii* was found in the western Caucasus as well [25,26]. According to authors “*Ae. squarrosa* (*Ae. tauschii*) grows in lowlands and mountain foothills, rarely in dry and humid silty areas up to the middle of the mountain, mainly in desert, semi-desert and field vegetation, as well as weeds in Abkhazia, Samegrelo, Imereti, Guria, Adjara, Kartli, in outer Kakheti, Gardabani” [25]. It can be assumed that two *Ae. tauschii* (A) and (B) participated in the formation of hexaploids through the D genome: *Ae. tauschii* (A)—macha (1–5, 7, 8, 10–12)—and *Ae. tauschii* (B)—macha M6, M9, CS, RD (Table 3 and Table 7).

*T. aestivum* subsp. *catrhlicoides* most likely originated in western Transcaucasia [3]; In 1967 Kuckuck had found populations with *carthlicium* and *carthlicoides* types near the border of East Turkey (Ardahan and Kars Provinces, Turkey) and West Georgia. This hexaploid wheat accession showed the subsp. *carthlicum*-like morphology. Subsp. *carthlicum* was proposed to have originated from spontaneous hybridization between subsp. *carthlicoides* and cultivated emmer wheat, *T. turgidum* subsp. *dicoccon* (Schrank) Thell. Subsp. *carthlicoides* should be considered as the original and elder genotype from which genes for this particular morphology of the ear were transferred together with the Q-factor to *T. carthlicum* [3,18]. According to Kuckuck, this region was distinguished by a tremendous genetic variation in wheat including *T. dicoccum, T. carthlicum* and *T. aestivum* subsp. *macha* [3]. It can be assumed that subsp. *carthlicoides* took part in the formation of the subsp. *macha* as an ancestor.

One of the four subspecies of wheat detected in Georgia is hexaploid, domesticated, hulled wheat *T. aestivum* subsp. *macha* [15]. This subspecies was detected in West Georgia in 1928 and described by Dekaprelevich and Menabde [27]. It is endemic to Georgia and is cultivated along with tetraploid West Georgian wheat (*T. turgidum* subsp. *palaeocolchicum*) [28].

It is proposed that macha and West Georgian wheats are sibling cultivars that arose in a hybrid swarm involving *T. aestivum* and wild emmer wheat [29]. It is accepted that spelt wheat is derived from free-threshing hexaploid wheat (*T. aestivum* subsp. *aestivum*) by hybridization with hulled emmer (*T. turgidum*). Experimental data suggest that European and Asian spelt may be polyphyletic. Free-threshing, hexaploid wheat seems to precede spelt. At least some European spelt originated from hybridization of club wheat (*T. aestivum* subsp. *compactum*) with emmer (domesticated hulled *Triticum turgidum*). Free-threshing hexaploid wheat was an ancestor of not only European spelt but also of some of the Asian forms of spelt, although the exact role free-threshing wheat has played is debatable [1].

The comparative analysis of complete nucleotide sequence 20 samples of hexaploid wheats offers the opportunity to draw several conclusions: 20 hexaploid wheat samples are divided into two groups—*Triticum aestivum* subsp. *spelt**a* three samples + *T. aestivum* subsp. *vavilovii* (Vav1) (collected in Armenia), and the remaining 16 samples including *T. aestivum* subsp. *vavilovii* (Vav2) (Collected in Sweden).

Hirosawa et al. [30] found two cpDNA SSR lineages in *T. aestivum*: Plastogroup I and II. Most subsp. *aestivum*, *compactum*, *macha*, and *sphaerococcum* accessions belonged to Plastogroup I. Subspecies *spelta* was split into Plastogroup I (70% of accessions examined) and Plastogroup II (30%). So, given the numbers of subsp. *spelta* accessions used (27 accessions in Hirosawa et al. and 3 in our work), the cpDNA trees (Hirosawa et al.’s and ours) appear quite consistent. Plastogroup I (and our major lineage) might represent a major lineage that was transmitted from cultivated *T. turgidum* via hybrid cross with *Ae. tauschii* and its subsequent allopolyploidization. Plastogroup II (and our minor lineage) might have resulted from introgression between hexaploid and tetraploid wheats, because subsp. *spelta* has the emmer wheat ancestry in Europe and Asia [1].

PCR-based haplotype analysis of the fourth intron of *Wknox1d* and the fifth-to-sixth-exon region of *Wknox1b* of all 20 hexaploid wheat samples was carried out. PCR-based haplotype analysis of the fourth intron of *Wknox1d* and the fifth-to-sixth-exon region of *Wknox1b* provides an opportunity to make an assumption that hexaploid wheats *T. aestivum* subsp. *macha* var. *palaeocolchicum* and var. *letshckumicum* differ from other macha samples by the absence of a 42 bp insertion in the fourth intron of *Wknox1d*. It can be assumed that two *Ae. tauschii* (A) and (B) participated in the formation of hexaploids through the D genome: *Ae. tauschii* (A)—macha (1–5, 7,8,10–12)—and *Ae. tauschii* (B)—macha M6, M9, *T. aestivum* subsp. *aestivum* cv. ‘Chinese Spring’ and cv. ‘Red Doly’. Another possible explanation would be that the insertion might have arisen during the diversification process of common wheat after polyploidization. 

In recent years, wheat yields have not increased per hectare, which is important due to the growing world population. Transferring agronomically important genes from wild relatives to common wheat has been shown to be an effective genetic resource for hexaploid wheat improvement. Advances in new technologies have made the complete wheat reference genome available, which offers a promising future for the study of wheat improvement which is essential to meet global food demand [31].

## 4. Materials and Methods

### 4.1. Plant Material, DNA Isolation, PCR Analysis, Genomic DNA Library Preparation and Sequencing on an Illumina NovaSeq 6000 Platform

The seeds of hexaploid wheat samples were received from the seed bank of The Agricultural University of Georgia (the late Prof. P. Naskidashvili) and Institute of Botany (Ilia State University, Tbilisi, Georgia) (ISU); The U.S. National Plant Germplasm System (GRINGlobal); The National BioResource Project—WHEAT Centre (Graduate School of Agriculture, Kyoto University, Kyoto, Japan); The Scientific Research Center of Agriculture (SRCA) (Georgia) and Tel Aviv University (Israel). The seeds were germinated in water at room temperature. Total genomic DNA extraction from young leaves, the construction of genomic DNA libraries and assembly of cpDNA have been described earlier [15,31]. An automatic annotation of cpDNA was performed by CpGAVAS [32]. For detection of SNPs (single nucleotide polymorphism) and Indels (insertion/deletion) and phylogeny tree construction, computer programs Mafft and Blast were used [33,34].

### 4.2. Construction of Shotgun Genomic DNA Libraries

Construction of 13 shotgun genomic libraries and sequencing on the NovaSeq 6000 was carried out at the Roy J. Carver Biotechnology Center, University of Illinois at Urbana-Champaign (UIUC). The shotgun genomic DNA libraries were constructed from 50 ng of DNA after sonication with a Covaris ME220 (Covaris, MA, USA) to an average fragment size of 400 bp with the Hyper Library Preparation Kit from Kapa Biosystems (Roche, CA, USA). To prevent index switching, the libraries were constructed using unique dual-indexed adaptors from Illumina (San Diego, CA, USA). The individually barcoded libraries were amplified with 6 cycles of PCR and run on a Fragment Analyzer (Agilent, CA, USA) to confirm the absence of free primers and primer dimers and to confirm the presence of DNA of the expected size range. Libraries were pooled in equimolar concentration and the pool was further quantitated by qPCR on a BioRad CFX Connect Real-Time System (Bio-Rad Laboratories, Inc., Hercules, CA, USA).

### 4.3. Sequencing of Libraries in the NovaSeq

The pooled, barcoded, shotgun libraries were loaded on a NovaSeq SP lane for cluster formation and were sequenced for 150 cycles from each side of the DNA fragments. The fastq read files were generated and demultiplexed with the bcl2fastq v2.20 Conversion Software (Illumina, San Diego, CA, USA).

### 4.4. PCR Analysis of Three Homoeologous Wheat Wknox1 Gene 

The genomic sequences of the *Wknox1d* 4th intron regions were amplified by PCR using the primer pair of Takumi a. Morimoto [18]: 5′-AAAAAAAAGGTTAAATGGAC-3′ and 5′-ACCTTATACATGATTGGGAA-3′. The *Wknox1b* 5th-to-6th exon region was amplified by PCR using the primer pair 5′-GCTGAAGCACCATCTCCTGA-3′ and 5′-CATGTAGAAGGCGGCGTTAG-3′. The DNA fragments (PCR products) were excised from the agarose gel. The DNA extraction and purification from the agarose gel was performed by the QIAquick Gel Extraction Kit (QIAGEN). PCR products were purified with GenElute PCR Clean-Up Kit (Sigma-Aldrich), dye labeled using a Big Dye Terminator Kit (Applied Biosystems), and sequenced on Applied Biosystems 3700 genetic analyzer (Laboratory Services Division of the University of Guelph, Guelph, ON, Canada).

## 5. Conclusions

Based on complete chloroplast DNA sequences, the 20 hexaploid wheat samples can be divided into two groups—*T. aestivum* subsp. *spelta* three samples + *T. aestivum* subsp. *vavilovii* collected in Armenia, and the remaining 16 samples, including *T. aestivum* subsp. *vavilovii* collected in Europe (Sweden).

Based on the fourth intron of *Wknox1d* and the fifth-to-sixth-exon region of *Wknox1b* hexaploid wheats, *T. aestivum* subsp. *macha* var. *palaeocolchicum* and var. *letshckumicum* were found to differ from other macha samples by the absence of a 42 bp insertion in the fourth intron of *Wknox1d*.

Two *Aegilops tauschii* Coss. (A) and (B) participated in the formation of hexaploids through the D genome: *Ae. tauschii* (A)—macha (1–5, 7, 8, 10–12)—and *Ae. tauschii* (B)—macha M6, M9, *T. aestivum* subsp. *aestivum* cv. ‘Chinese Spring’ and cv. ‘Red Doly’.

## Figures and Tables

**Figure 1 ijms-22-12723-f001:**
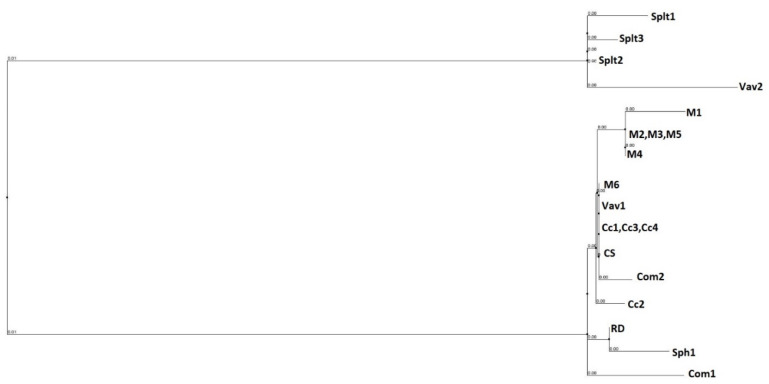
Complete chloroplast genome phylogeny of hexaploid wheats.

**Figure 2 ijms-22-12723-f002:**
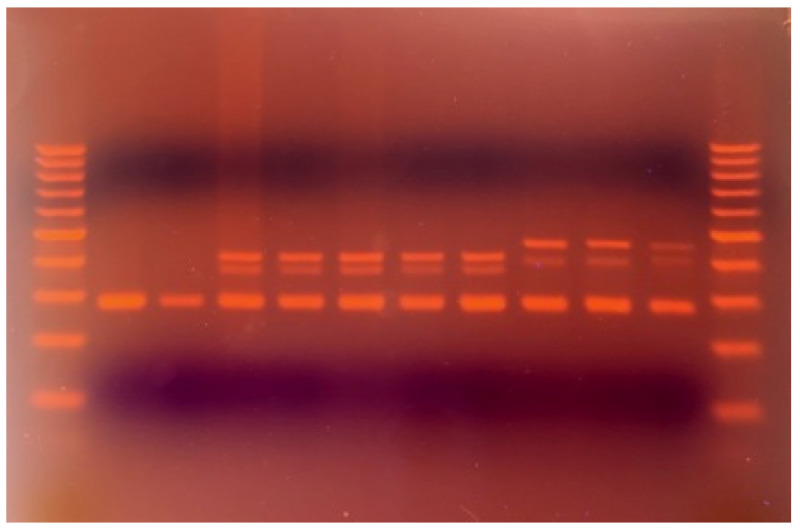
Image of 2% agarose gel electrophoresis of PCR-amplified fourth intron of *Wknox1d* DNA region of hexaploid wheat. Lanes: 2—*T. turgidum* subsp*. carthlicum*, 3—*T. turgidum* subsp. *durum* (Ldn), 4—*T. aestivum* subsp. *aestivum* (CS), 5—*T. aestivum* subsp. *carthlicoides*, 6—*T. aestivum* subsp. *carthlicoides*, 7—*T. aestivum* subsp. *macha* (M_6), 8—*T. aestivum* subsp*. macha* (M_9), 9—*T. aestivum* subsp. *macha* (M_1), 10—*T. aestivum* subsp. *macha* (M_2), 11—*T. aestivum* subsp. *macha* (M_3); Lanes 1, 12—100 bp DNA marker.

**Table 1 ijms-22-12723-t001:** List of hexaploid wheats.

	Species	Genome	Plasmon
Hulled *T. zhukovskyi*		GGA^u^A^u^A^m^A^m^	G
Hulled *T. aestivum*			
	*T. aestivum* subsp. *spelta* Thell.	BBA^u^A^u^DD	B
	T. *aestivum* subsp. *macha* (Dekapr. and Menabde) Mackey		
	*T. aestivum* subsp. *vavilovii* Jakubz.		
Free-threshing *T. aestivum*	*T. aestivum* subsp. *aestivum*		
	*T. aestivum* subsp. *compactum* (Host) Mackey		
	*T. aestivum* subsp. *sphaerococcum* (Percival) Mackey		
	*T. aestivum* subsp. *carthlicoides* nom. nud.		

**Table 2 ijms-22-12723-t002:** Jacubciner’s classification system (1958) According to Yen a. Yang [2].

Congretio	Species	Distribution Areas	Habits
Hexaploidea	Cultivated hulled		
2n = 42	*T. zhukovskyi* Men.et Er.	Georgia	Springness
	*T. macha* Dek.et Men.	Georgia	Winterness
	*T. spelta* L.	Iran, south Germany, Spain	Winterness, Springness
	Cultivated naked grain		
	*T. aestivum* L.	All over the world	Springness, winterness, half winterness
	*T. compactum* Host	Transcaucasia, Kazakhstan, Asia Minor, Afghan, Chile	Springness, winterness, half-winterness
	*T. vavilovii* Jakubz.	Armenia	Winterness
	*T. sphaerococcum* Perc.	Pakistan, India	Springness

**Table 3 ijms-22-12723-t003:** Hexaploid wheat accessions used in the study.

	Botanical Name	GenBank Accession Number	
Hulled			
M1	*Triticum aestivum* L. subsp. *macha* (Dekapr. and Menabde) Mackey var. *megrelicum* (Menabde)	LC372826	ISU
M2	*Triticum aestivum* L. subsp. *macha* (Dekapr. and Menabde) Mackey var. *georgicum* (Menabde)	LC373211	ISU
M3	*Triticum aestivum* L. subsp. *macha* (Dekapr. and Menabde) Mackey var. *colchicum* (Dekapr. and Menabde)	LC375536	ISU
M4	*Triticum aestivum* L. subsp. *macha* (Dekapr. and Menabde) Mackey var. *scharaschidzei* (Menabde)	LC374397	ISU
M5	*Triticum aestivum* L. subsp. *macha* (Dekapr. and Menabde) Mackey var. *palaeoimereticum* (Dekapr. and Menabde)	LC375773	ISU
M6	*Triticum aestivum* L. subsp. *macha* (Dekapr. and Menabde) Mackey var. *palaeocolchicum* (Dekapr. and Menabde)	NC_025955	ISU
M7	*Triticum aestivum* L. subsp. *macha* (Dekapr. and Menabde) Mackey var. *ericzjanae* Menabde		ISU
M8	*Triticum aestivum* L. subsp. *macha* (Dekapr. and Menabde) Mackey var. *ibericum* Dekapr. and Menabde		ISU
M9	*Triticum aestivum* L. subsp. *macha* (Dekapr. and Menabde) Mackey var. *letshchumicum* Dekapr. and Menabde		ISU
M10	*Triticum aestivum* L. subsp. *macha* (Dekapr. and Menabde) Mackey var. *subcolchicum* Dekapr. and Menabde		ISU
M11	*Triticum aestivum* L. subsp. *macha* (Dekapr. and Menabde) Mackey var. *submegrelicum* Dekapr. and Menabde		ISU
M12	*Triticum aestivum* L. subsp. *macha* (Dekapr. and Menabde) Mackey var. *subletshchumicum* Dekapr. and Menabde		ISU
Splt2	*Triticum aestivum* L. subsp. *spelta* (L.) Thell.	LC625352	PI 348000
Splt1	*Triticum aestivum* L. subsp. *spelta* (L.) Thell.	LC625353	PI 348220
Splt3	*Triticum aestivum* L. subsp. *spelta* (L.) Thell.	LC625866	PI 191393
Vav1	*Triticum vavilovii* Jakubz.	LC621349	PI 326319
Vav2	*Triticum vavilovii* Jakubz.	LC625865	PI 428342
Free-threshing			
RD	*Triticum aestivum* L. subsp. *aestivum* var. *ferrugineum* (Alef.) Mansf. cv. ‘Red Doly’	LC377169	ISU
CS	*Triticum aestivum* L. subsp. *aestivum* cv. ‘Chinese Spring’	LC622404	
Cc1	*Triticum aestivum* L. subsp. *carthlicoides* nom. nud.	LC621350	PI 262678
Cc3	*Triticum aestivum* L. subsp. *carthlicoides* nom. nud.	LC621195	SRCA
Cc2	*Triticum aestivum* L. subsp. *carthlicoides* nom. nud.	LC622405	PI 532901
Cc4	*Triticum aestivum* L. subsp. *carthlicoides* nom. nud.	LC621194	SRCA
Com2	*Triticum aestivum* L. subsp. *compactum* (Host) Mackey	LC623766	KU-9873
Com1	*Triticum aestivum* L. subsp. *compactum* (Host) Mackey	LC623764	ISU
Sph1	*Triticum aestivum* L. subsp. *sphaerococcum* (Percival) Mackey	LC623765	Cltr17737

**Table 4 ijms-22-12723-t004:** SNPs specific for chloroplast DNA of hexaploid wheats.

Nucleotide Position According to CS	Locus	CS	M6	Vav1	Cc2	Cc1, Cc3, Cc4	RD	Sph1	M2, M3, M5	M4	M1	Com1	Com2	Splt1	Splt2	Splt3	Vav2	Amino Acid Substitution
2903	Gene *matK*	G											C					A–G
3400	Intergenic *matK-trnK-UUU*	C												A	A	A	A	
4918	Intron *rps16*	T														C		
13,015	Intergenic *trnfM-CAT-trnT*-GGT	T												C	C	C	C	
14,580	Intergenic *trnfM-CAT-trnT*-GGT	C						G										
15,371	Intergenic *trnY-GTA-trnD-GTC*	A												G	G	G	G	
16,481	Intergenic *trnD-GTC-psbM*	C												T	T	T	T	
20,853	Gene *rpoB*	C												T	T	T	T	Syn
25,444	Gene *rpoC2*	C												A	A	A	A	Syn
32,710	Intergenic *atpH-atpF*	C						T										
44,509	Intergenic *ycf3 trnS-GGA*	C			A													
45,856	Intergenic *rps4-trnT-TGT*	A												C	C	C	C	
46,434	Intergenic *trnT-TGT trnF-GAA*	G												T	T	T	T	
47,628	Intergenic *trnT-TGT-trnF-GAA*	G					A	A						G	G	G	G	
50,332	Intergenic *ndhC-trnM-CAT*	G												T	T	T	T	
50,716	Intergenic *ndhC-trnM-CAT*	T												C	C	C	C	
52,295	Gene *atpB*	G												T	T	T	T	Q–K
56,472	Intergenic *rbcL-psaI*	A									G			G	G	G	G	
56,670	Intergenic *rbcL-psaI*	C												T	T	T	T	
63,289	Gene *petG*	A												T	T	T	T	Syn
64,318	Intergenic *psaJ-rpl33*	C												T	T	T	T	
72,207	Intergenic *petB-petD*	C												T	T	T	T	
73,742	Intergenic *petD-rpoA*	T												C	C	C	C	
78,113	Intron *rpl16*	C												T	T	T	T	
78,732	Intergenic *rpl16-rps3*	G												T	T	T	T	
82,499	Intergenic *rpl23-trnI-CAT*	T										G						
100,961	Gene *rps15*	T									G							Syn
102,561	Gene *ndhF*	C							T	T	T							Syn
104,993	Intergenic *rpl32-trnL-TAG*	A												C	C	C	C	
105,477	Intergenic *rpl32-trnL-TAG*	C												A	A	A	A	
106,672	Gene *ccsA*	C												T	T	T	T	Syn
113,487	Gene *ndhH*	T															G	Syn
114,943	Gene *rps15*	A									C							F–L
133,405	Intergenic *trnI-CAT-rpl23*	A										C						

**Table 5 ijms-22-12723-t005:** Indels specific for chloroplast DNA of hexaploid wheats.

Nucleotide Position According to CS	Locus	CS	M6	Vav1	Cc2	Cc1, Cc3, Cc4	RD	Sph1	M2, M3, M5	M4	M1	Com2	Com1	Splt1	Splt2	Splt3	Vav2
4175	Intergenic *matK-rps16*	-												AAAAT	AAAAT	AAAAT	AAAAT
7422	Intergenic *psbI-trnS-GCT*	15T												18T	12T	12T	12T
8219	Intergenic *trnS-GCT-psbD*	10T					11T										
8407	Intergenic *trnS-GCT-psbD*	-										TATTTCT		ATTTCT	ATTTCT	ATTTCT	ATTTCT
11,183	Intergenic *psbC-trnS-TGA*	14A										13A		13A	13A	13A	
17,991	Intergenic *petN-trnC-GCA*	10A	11A				11A					11A		11A	11A	11A	11A
18,757	Intergenic *trnC-GCA-rpoB*	19C												9C	8C	8C	
31,609	Intergenic *atpI-atpH*	12T								11T							
32,442	Intergenic *atpH-atpF*													+A	+A	+A	+A
33,718	*Intron* *atpH*	14A										13A		13A	13A	13A	12A
42,920	*I intron ycf3*	10T													9T	9T	9T
42,931	*I intron ycf3*	3T												2T			
45,993	Intergenic *rps4-trnT-TGT*	8A												9A	9A	9A	9A
47,788	Intergenic *trnF-GAA-ndhJ*	15A								14A				11A	11A	11A	10A
56,724	Intergenic *rbcL-psaI*	10T															9T
60,767	Intergenic *petA psbJ*													+CATT	+CATT	+CATT	+CATT
60,773	Intergenic *petA-psbJ*	17T	16T	16T	16T	16T	16T		16T	16T	16T	16T	15T				16T
62,041	Intergenic *psbE-petL*	-												+TA	+TA	+TA	+TA
Duplication_70,856	Intron *petB*													TATT	TATT	TATT	TATT
71,219	Intron *petB*	6T												7T	7T	7T	7T
73,586	Intergenic *petD-rpoA*	5T												6T	6T	6T	6T
75,699	Intergenic *rpl36-infA*	10A												11A	11A	11A	
76,051-62_duplication	*Intergenic* infA-rps8							CTGTCATATTTT									
76,599	Intergenic *rps8-rpl14*	10T					11T		11T	11T				9T	9T	9T	
77,140	Intergenic *rpl14-rpl16*	10T											9T	9T	9T	9T	9T
86,581	*Intron* *ndhB*	5T															4T
104,113	Intergenic *ndhF-rpl32*	13A												11A	12A	14A	
129,319	*Intron* *ndhB*	5A															4A

**Table 6 ijms-22-12723-t006:** Inversions and a long insertion specific for chloroplast DNA of hexaploid wheats.

Nucleotide Position According to CS	Locus	Splt1, Splt2, Splt3, Vav2		Length
79,532	Gene *rpl22*	GATGGATCTAAAGGTTATTTAGATTTCTTTACTAT	Insertion	35 bp
105,139–105,196	Intergenic *rpl32—**trnL-TAG*	ACTTTTCATAATTTTCATAATAGAATCCT CATATTTTATTATGAAAATTATGAAAAGT	Inversion with 14 bp loop	58 bp
106,795–106,819	Intergenic *ccsA—ndhD*	AAAACCTTCATGAAATGAAGGTTTT	Inversion with 3 bp loop	25 bp
135,896-52	Intergenic *rps19—psbA*	AAAGACAGAAATACCCAATATCTTGCTA GAACAAGATATTGGGTATTTCTGTCTTT	Inversion with 6 bp loop	56 bp

**Table 7 ijms-22-12723-t007:** PCR-based haplotype analysis of the fourth intron of *Wknox1d*. A 122-bp MITE insertion specifically observed at the D-genome locus of *Wknox1* generates the upper bands (411 bp) in hexaploid wheat accessions (except *Triticum aestivum* L. subsp. *compactum*) and 453 bp in macha accessions.

Sample	PCR Product, bp				
	277, 284	375	400	411	453
*Triticum aestivum* subsp.*aestivum* cv. ‘Chinese Spring’ (CS)	+	+		+	
*Triticum aestivum* subsp. *aestivum* var. *ferrugineum* (Alef.) Mansf. cv. ‘Red Doly’ (RD)	+	+		+	
*Triticum aestivum* subsp. *carthlicoides* (Cc1, Cc2. Cc3. Cc4)	+	+		+	
*Triticum aestivum* subsp. *compactum* (Com2)	+				
*Triticum aestivum* subsp. *compactum* (Com1)	+	+		+	
*Triticum aestivum* subsp. *sphaerococcum* (Sph1)	+	+		+	
*Triticum aestivum* subsp. *macha* (M6, M9)	+	+		+	
*Triticum aestivum* subsp. *macha* (M1, M2, M3, M4, M5, M7, M8 M10, M11, M12)	+		+		+
*Triticum aestivum* subsp. *spelta* (splt1, splt2, splt3)	+	+		+	
*Triticum vavilovii* (Arm) (Vav1)	+	+		+	
*Triticum vavilovii* (Eur) (Vav2)	+	+		+	
*Triticum turgidum* subsp. *carthlicum* var. *rubiginosum*	+				
*Triticum turgidum* subsp. *durum* cv. ‘Langdon’ (Ldn)	+				

**Table 8 ijms-22-12723-t008:** PCR-based haplotype analysis of the fifth-to-sixth exon region of *Wknox1b*.

Sample	PCR Product, bp			
	410	429, 430	560	588
*Triticum aestivum* subsp.*aestivum* cv. ‘Chinese Spring’ (CS)		+	+	+
*Triticum aestivum* subsp. *aestivum* var. *ferrugineum* cv. ‘Red Doly’ (RD)		+	+	+
*Triticum aestivum* subsp. *carthlicoides* (Cc1, Cc2, Cc3, Cc4)		+		
*Triticum aestivum* subsp. *compactum* (Com2)		+	+	+
*Triticum aestivum* subsp. *compactum* (Com1)		+		
*Triticum aestivum* subsp. *sphaerococcum* (Sph1)		+	+	+
*Triticum aestivum* subsp. *macha* (M6, M9)		+		
*Triticum aestivum* subsp. *macha* (M1, M2, M3, M4, M5, M7, M 8, M10, M11, M12)		+		
*Triticum aestivum* subsp. *spelta* (splt1, splt2, splt3)		+	+	+
*Triticum vavilovii* (Arm) (Vav1)	+	+		
*Triticum vavilovii* (Eur) (Vav2)		+	+	+
*Triticum turgidum* subsp. *carthlicum* var. *rubiginosum*		+		
*Triticum turgidum* subsp. *durum* cv. ‘Langdon’ (Ldn)		+	+	+

**Table 9 ijms-22-12723-t009:** Hexaploid wheats with matching values of fourth intron of *Wknox1d* and fifth-to-sixth exon region of *Wknox1b*.

*Triticum aestivum* subsp. *carthlicoides* (Cc1, Cc2, Cc3, Cc4)
*Triticum aestivum* subsp. *compactum* (Host) (Com1)
*Triticum aestivum* subsp. *macha* (M6, M9)
*Triticum vavilovii* (Arm) (Vav1)

## Data Availability

The complete chloroplast DNA sequence data supporting this study are openly available in GenBank at https://www.ncbi.nlm.nih.gov/nuccore/LC625352,LC625353,LC625866,LC621349,LC625865,LC377169,LC622404,LC623765,LC621350,LC623764,LC621194,LC622405,LC623766 (accessed on 8 April 2021).

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
