# Peer review of "Genetic Analysis of Hexaploid Wheat (Triticum aestivum L.) Using the Complete Sequencing of Chloroplast DNA and Haplotype Analysis of the Wknox1 Gene"

_ijms, 2021, doi:10.3390/ijms222312723_

Round 1
Reviewer 1 Report
Congratulation for this paper!
Author Response
No comments
Reviewer 2 Report
Name of Journal: International Journal of Molecular sciences
Manuscript ID: ijms-1384148
Title of the article: Genetic Analysis of Hexaploid Wheat (Triticum aestivum L.)
The paper is original; it reads well and is of importance as mapping of quantitative trait loci determining agronomic important characters in hexaploid wheat (Triticum aestivum L.). Many authors reported that the total biomass of the hexaploid genotypes was 36% greater than of the diploid genotypes.
The paper is well written and there is no critical comments except very few language mistakes mentioned below:
Abstract
Line 9: genetic Correct to a genetic
Line 11: 4th Correct to 4th
Line 11: the 5th-to-6th Correct to 5th –to-6th
Line 21: 4th Correct to 4th
Line 22: provide Correct to provides
Line 23: actually Delete this word
Introduction
Line 38: include Correct to includes
Line 71: One of the possibilities to Correct to One possibility to
Line 77: appropriate Delete this word
Line 83: are Correct to is
Line 85: a large number of Correct to numerous
Results
Line 190: by Correct to into
Line 197: investigation Correct to investigation,
Line 201: agarose gel Correct to an agarose gel
Figure 2: 100 bp should refer by arrow on Lanes 1 and 12 of the photo for clarification
Discussion
Line 268: participated Correct to took part
Line 284: of Delete
Line 288: actually Delete this word
The authors should discuss the following:
- Genetic diversity using reliable methods provides useful information for the management of genetic resources and crop improvement programs.
- Importance of hexaploid wheats for yield increase and biomass or resistance to pathogens.
- The genetic diversity in wild relatives of wheat.
Materials and Methods
Line 320: concentration and Correct to concentration, and
Author Response
ANSVERS TO THE REVIEWER
- Genetic diversity using reliable methods provides useful information for the management of genetic resources and crop improvement programs.
- Importance of hexaploid wheats for yield increase and biomass or resistance to pathogens. The genetic diversity in wild relatives of wheat
All language mistakes are corrected.
One paragraph concerning biomass and hexapoidicity is added.
Reviewer 3 Report
The work of Gogniashvili et al. deals with the genetic characterization of hexaploidy wheat Triticum aestivum L by using two approaches – complete sequencing of the chloroplast DNA, and a nuclear genetic marker – the 4th intron of the Wknox1d and the 5-6 exon of the Wknox1b, of a totally 20 samples (13 in the present study and by using the information of additional 7, published previously by the authors’ team). The methodology is represented in a reproducible manner while the results are of scientific interest. However, I would advise some major revisions:
- Title: The authors should include in the title something which indicates that their study includes typical kinds of wheat for Georgia and/or the Caucasus.
- Abstract: 1) The results from the chloroplast sequencing should be included; and 2) again it should be indicated that the main motivation of this research is the study of the regional kinds of wheat.
- Results: The content of the two paragraphs from line 120 to line 133 represents a discussion and not results.
- Discussion: 1) No discussion of the chloroplast sequencings is present; 2) The discussion of the results of the Wknox gene is rather scarce and should be extended; and 3) The content of the paragraphs from line 246 to line 282 which consist of 80% of the entire section is not directly related to the obtained results.
Although the manuscript is very interesting, it could not be published in its current form.
Author Response
ANSVERS TO THE REVIEWER
- Title: The authors should include in the title something which indicates that their study includes typical kinds of wheat for Georgia and/or the Caucasus.
- Abstract: 1) The results from the chloroplast sequencing should be included; and 2) again it should be indicated that the main motivation of this research is the study of the regional kinds of wheat.
We completely disagree with the reviewer in this judgment. The article is dedicated to Hexaploid wheat in general and not to Georgia and the Caucasus. Of the sequenced chloroplast DNA of 13 samples belonging to 6 subspecies of hexaploid wheat, none is Georgian.
1.T. aestivum subsp. aestivum;
2.T. aestivum subsp. compactum;
3.T. aestivum subsp. sphaerococcum;
4.T. aestivum subsp. carthlicoides.
- T. aestivum subsp. spelta;
6.T. aestivum subsp. vavilovii jakubz.
Georgian subspecies of Triticum are:
1 T. aestivum subsp. macha
2 T. zhukovskyi Menabde & Ericzjan
3. Results: The content of the two paragraphs from line 120 to line 133 represents a discussion and not results.
4. Discussion: 1) No discussion of the chloroplast sequencings is present; 2) The discussion of the results of the Wknox gene is rather scarce and should be extended; and 3) The content of the paragraphs from line 246 to line 282 which consist of 80% of the entire section is not directly related to the obtained results.
Rearranged in the results and discussion section
Discussion of the chloroplast sequences are added.
Round 2
Reviewer 3 Report
Dear authors and Editor,
Although the authors agreed with remarks #3 and #4 of my previous revision, no significant changes were made regarding my previous remarks #1 and #2, neighther they were addressed. They were:
- Title: The authors should include in the title something which indicates that their study includes typical kinds of wheat for Georgia and/or the Caucasus.
- Abstract: 1) The results from the chloroplast sequencing should be included; and 2) again it should be indicated that the main motivation of this research is the study of the regional kinds of wheat.
That is why unfortunately I cannot accept the manuscript in its current form and recommend to be revised.
Author Response
Answer to the reviewer
- The title of the manuscript was changed:
Genetic analysis of hexaploid wheat (Triticum aestivum L.) using the complete sequencing of chloroplast DNA and haplotype analysis of the Wknox1 gene
- Abstract was changed:
a/The results from the chloroplast sequencing was included;
b/ the regional kinds of wheat were added.
Round 3
Reviewer 3 Report
I still continue to insist that the chloroplast results should be mentionned within the abstract.
Author Response
The chloroplast results was added to the abstracts - 105 words.
